# Metacommunity analyses show an increase in ecological specialisation throughout the Ediacaran period

**Rebecca Eden, Andrea Manica📧, Emily G. Mitchell📧***

Department of Zoology, University of Cambridge, Cambridge, United Kingdom

* ek338@cam.ac.uk

**Data Availability Statement:** All data and code is available in the supplementary materials. The data used in this paper has been modified from previously published data and are publicly available on figshare doi: 10.6084/m9.figshare.13664105.

## Abstract

The first animals appear during the late Ediacaran (572 to 541 Ma); an initial diversity increase was followed reduction in diversity, often interpreted as catastrophic mass extinction. We investigate Ediacaran ecosystem structure changes over this time period using the "Elements of Metacommunity Structure" framework to assess whether this diversity reduction in the Nama was likely caused by an external mass extinction, or internal metacommunity restructuring. The oldest metacommunity was characterised by taxa with wide environmental tolerances, and limited specialisation or intertaxa associations. Structuring increased in the second oldest metacommunity, with groups of taxa sharing synchronous responses to environmental gradients, aggregating into distinct communities. This pattern strengthened in the youngest metacommunity, with communities showing strong environmental segregation and depth structure. Thus, metacommunity structure increased in complexity, with increased specialisation and resulting in competitive exclusion, not a catastrophic environmental disaster, leading to diversity loss in the terminal Ediacaran. These results reveal that the complex eco-evolutionary dynamics associated with Cambrian diversification were established in the Ediacaran.

## Introduction

One of the most dramatic events in the history of Earth is the sudden appearance of animals in the fossil record during the Ediacaran period (635 to 541 Ma), after billions of years of microbial life [1–3]. Ediacaran anatomies are particularly difficult to compare to modern phyla, which has hampered our understanding of Ediacaran evolution and how Ediacaran organisms relate to the Cambrian Explosion and extant animal phyla [4]. Patterns of taxonomic, morphological, and ecospace diversity change dramatically during the Ediacaran [5,6], which has led to the suggestion of several evolutionary radiations, corresponding to the Avalon, White Sea, and Nama assemblages [1,7–9]. These 3 assemblages consist of groupings of communities that occupy partially overlapping temporal intervals and water depths, with no significant litho-taphonomic or biogeographic influence [7,8,10]. The oldest assemblage, the Avalon (575 to 565 Ma), exhibits relatively limited ecological and morphological diversity [5,6], with only

**Funding:** This work was funded by a Natural Environment Research Council Independent Research Fellowship NE/S014756/1 to EGM. The funders had no role in study design, data collection and analysis, decision to publish, or preparation of the manuscript.

**Competing interests:** The authors have declared that no competing interests exist.

limited palaeoenvironmental influence on its composition and taxa interactions [11–14]. The White Sea assemblage (558 to 550 Ma) shows a large increase in morphological diversity, including putative bilaterians [5], in tandem with a greater ecological diversity that includes the appearance of grazing, herbivory, and widespread motility [15,16]. These innovations are coupled to the development of dense communities with high community heterogeneity between environments [16,17] and increased taxa sensitivity to fine-scale environment [12,18]. The Nama assemblage (549 to 543 Ma) includes the oldest biomineralising taxa and records a decrease in taxonomic diversity [5,19–21]. This reduction in taxonomic diversity, sometimes referred to as the "diversity drop," has been suggested to correspond to a post-White Sea extinction around 550 Ma, which eliminated the majority of Ediacaran soft-bodied organisms [9,22–24]. This diversity drop has been suggested to be caused by either an environmental driven catastrophic environmental extinction or biotic replacement driven extinction [9,21,23,24]. Recent work has shown that a biotic replacement driven extinction, whereby mobile metazoans outcompeted soft-bodied Ediacaran organisms through bioturbation and ecosystem engineering, is unlikely, due in part to prolonged co-occurrence of trace fossils with soft-bodied biota [1,25]. Other currently unknown and/or unpreserved intrinsic causes behind a biotic replacement model cannot be excluded at the moment.

Previous studies have focused primarily on defining the different assemblages and what the underlying factors behind the different assemblages [7,8,10], looking at taxonomic and morphological diversity between assemblages [5,26] with little investigation of how the ecological structure within the assemblages differs. The network structures of the co-occurrence of Ediacaran body fossil and trace fossil taxa were compared by Muscente and colleagues [9] who found compartmentalisation of the assemblages within the total Ediacaran network. However, the prior cluster analyses of [7,8,10] and network analyses of [9] have not assessed the relative frequency of taxa co-occurrences within assemblages; i.e., whether they were statistically different to what would be expected by random chance, nor compared the ecological structure within each assemblage to known ecological models.

In this study, we will investigate the structural attributes within these assemblages using 3 analyses that have not previous been used to investigate Ediacaran macroecology. We used presence–absence data encompassing 86 Ediacaran localities and 124 taxa, with paleoenvironment, depth, lithology, time, and assemblage data from [8,24] (S1 Fig). Ediacaran fossils are commonly found preserved in situ, so their bedding planes (the rock surfaces that preserve the fossils) preserve near-compete censuses of the communities [15,27]. This exceptional preservation means that ecological analyses normally reserved for modern communities can be applied (e.g., [12,28]).

First, we will use the "Elements of Metacommunity Structure" (EMS) framework to investigate emergent properties of groups of connected communities that may arise from taxa interactions, dispersal, environmental filtering, and the interaction of these factors [29–31] (Fig 1). Most fossil metacommunities do not fulfil the requirements of random sampling that would be needed to analyse them with such an ecological framework. However due to their exceptional preservation, the Ediacaran metacommunities are an exceptional census of the benthic assemblages present at the time, making them amenable to be analysed within the EMS framework. EMS does not assume even dispersal across all sites, with intermediate levels of disturbance associated with the highest levels of filtering of community by biotic and abiotic factors [29], and dispersal limitation associated with negative turnover [32]. Ediacaran communities vary in how much they are separated in time and space, from ecological to geological time scales [8,9,13], and their organisms have been shown to have large dispersal ranges based on reproductive mode [28,33,34] and species occurrence over large space and time scales [35]. Because the connectivity of these Ediacaran communities via dispersal has been established,

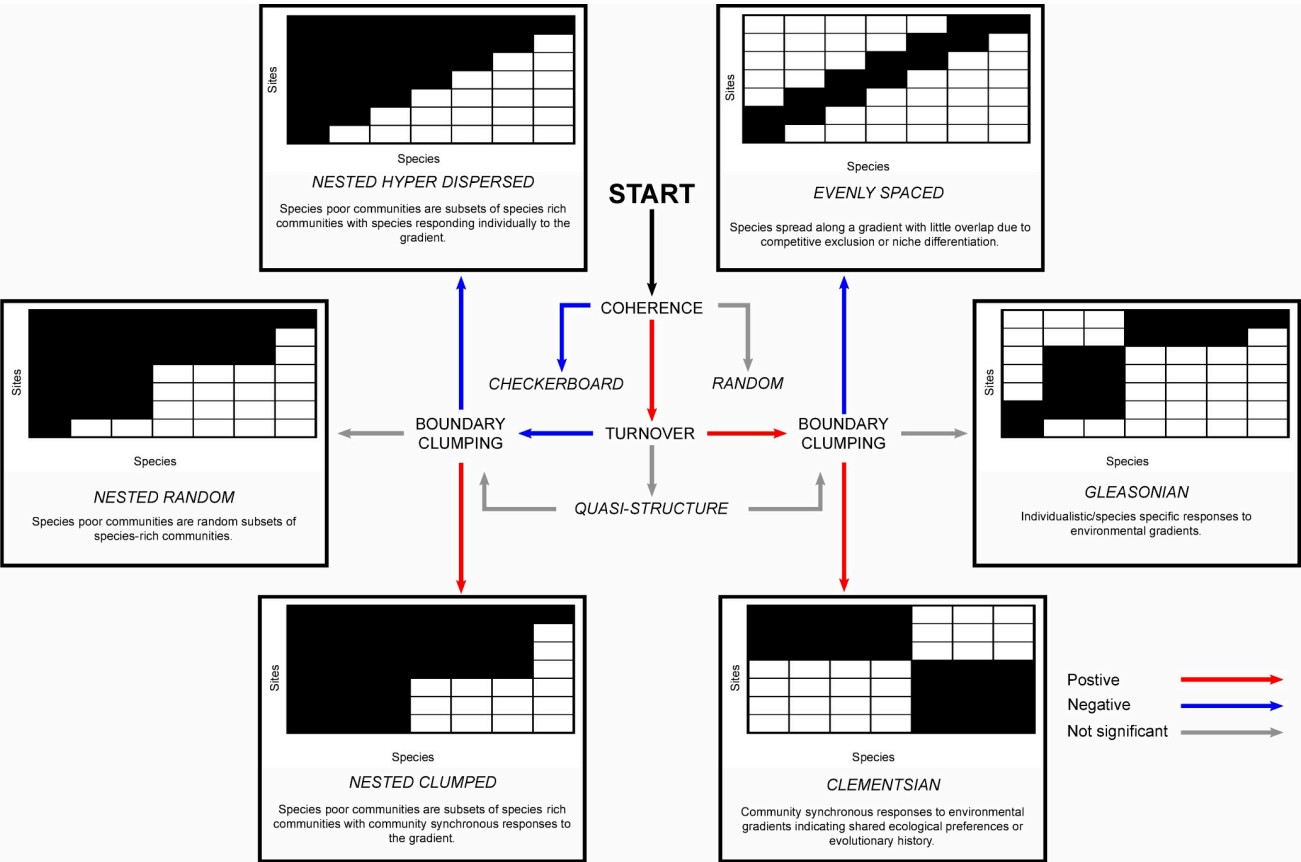

**Fig 1. Idealised metacommunity structures adapted from [30].** These graphs show taxa abundance patterns in idealised metacommunities of several taxa (represented by different colours), which respond to a latent environmental gradient (they exhibit significant positive coherence). The first step of the analyses (START) is to determine whether the metacommunity exhibits positive, negative, or random coherence. Random coherence corresponds to NS metacommunity structure; negative coherence is a checkerboard pattern [45], so significant mutual exclusivity between species and sites. Positive coherence indicates mutual co-occurring taxa associations, and there are several different possible models. For positive coherence, turnover and boundary clumping are calculated to determine the type of metacommunity structure. Nonsignificant turnover corresponds to quasi structures. These EMF analyses enable the structure of metacommunities to be grouped into one of 14 models: (1) random; (2) checkerboard; (3–5) nested clumped, random, and hyperdispersed; (6–8) Clementsian, Gleasonian, and evenly spaced; (9–10) Quasi nester clumped and hyperdispersed; and (11–12) quasi Clementsian and evenly spaced. See S1 Table. EMF, Elements of Metacommunity Framework; NS, no significant.

here we define metacommunities as sets of fossil localities (communities), which are connected by the dispersal of many species [29]. The EMS framework is a hierarchical analysis that identifies properties in site-by-taxa presence/absence matrices, which are related to the underlying processes shaping taxa distributions [31], but to date has limited application to the fossil record [36]. Three metacommunity metrics are calculated to determine the structure: coherence, turnover, and boundary clumping [29–31], which are hierarchical rather than independent of each other. The values and statistical significance of these metrics determine where the metacommunity fits within the 14 different metacommunity types within the EMS framework (Fig 1), with different metric combinations indicating different underlying processes behind the metacommunity structure. To determine whether an observed metric score differs significantly from random, we computed the z-score, which measures its distance from the mean of the randomisations (simulation mean) as the number of standard deviations (thus making it comparable across metrics with difference units). If the z-score is negative, the observed value is smaller than the simulated mean; if it is positive, then it is greater than the simulated mean; $z \geq 3$ indicates a significant deviation.

Coherence is a measure of the extent to which all the taxa respond to the same environmental gradient, where this gradient may result from the interplay of several biotic and abiotic factors that differ between sites [37]. Coherence is positive when the taxa in the site-by-taxa matrix all respond to the same environmental gradient. Most extant well-sampled metacommunities display significant positive coherence due to similarities in evolutionary history, ecological preferences, or life history trade-offs within communities [37]. A significant negative coherent site-by-taxa matrix reflects a high number of mutually exclusive taxa pairs creating checkerboard patterns [7,9,10]. These checkerboard patterns do not have further underlying structure (in contrast to positive coherence patterns; Fig 1), as there is no discernible gradient to which all the taxa respond. Negative co-occurrences and significant segregation/checkerboard patterns can be formed from strong competition, grazing/herbivory, or strongly non-overlapping niches, all of which form similar metacommunity patterns due the presence of mutually exclusive pairs of taxa [29,31,37]. A nonsignificant coherence reflects no significant metacommunity structuring (Fig 1). For metacommunities that have positive coherence, the turnover metric tests the amount of taxa replacement between sites [38]. If taxa ranges are nested within each other, there is less turnover than expected by chance along the gradient (significantly negative). If there are more differences in site taxa composition along the gradient than expected by chance, turnover is significantly positive and the structure is nonnested. Nonsignificant turnover indicates a weaker structuring mechanism, termed a quasi structure (Fig 1) [30]. Quasi structures have the same fundamental characteristics as the idealised structures, but because range turnover is not significantly different from random, it is likely that the underlying structuring mechanisms are weaker than those for which turnover is significant. The final metric, boundary clumping measures the extent to which taxa range limits cluster at the same sites across the environmental gradient [37]. The range limits can be clumped (significant positive), hyperdispersed (significant negative), or random (nonsignificant).

Positive coherence and negative turnover result in nested metacommunities with taxa-poor sites being predictable subsets of taxa-rich sites, implying that species are dispersal limited [39]. Nested metacommunities have been shown to be associated with a low degree of spatial connectivity and environmental variation [39] and have been shown to govern postextinction dynamics [40].

Clumped species boundaries tend to be associated with the transitions between different biomes, where 2 biological communities mix, in contrast to hyperdispersed species loss where species loss is evenly distributed across the range [30]. Positive coherence and turnover with hyperdispersed (negative) boundary clumping describes an evenly spaced metacommunity (Fig 1). Where coherence, turnover and boundary clumping are all positive, the metacommunity is classed as Clementsian (Fig 1), where groups of taxa with similar range boundaries co-occur and respond in a similar way to environment gradients [37,41]. Taxa within Clementsian metacommunities respond synchronously to environmental gradients, suggesting physiological or evolutionary trade-offs associated with environmental thresholds [42], tend to result from high levels of environmental variability and spatial connectivity [39], and are found to be the most common (e.g., [32]). When coherence and turnover are positive but there is no significant boundary clumping, the metacommunity is described as Gleasonian (Fig 1) where each taxon reacts individualistically to environmental gradients [30].

Secondly, we used Spearman rank correlations to test whether within-assemblage community composition is correlated with depth. The ordering of the sites was given by the ordination output from the EMS analyses (Fig 2), which is produced by reciprocal averaging, a type of correspondence analysis that ordinates the sites (y-axis of Fig 2) based on their species composition (x-axis of Fig 2) [31]. This ordering groups the sites together with similar community composition, and we can see from Fig 2 that the assemblages (indicated by different colours) are grouped together and that the depths (shown alongside the y-axis) show a correspondence

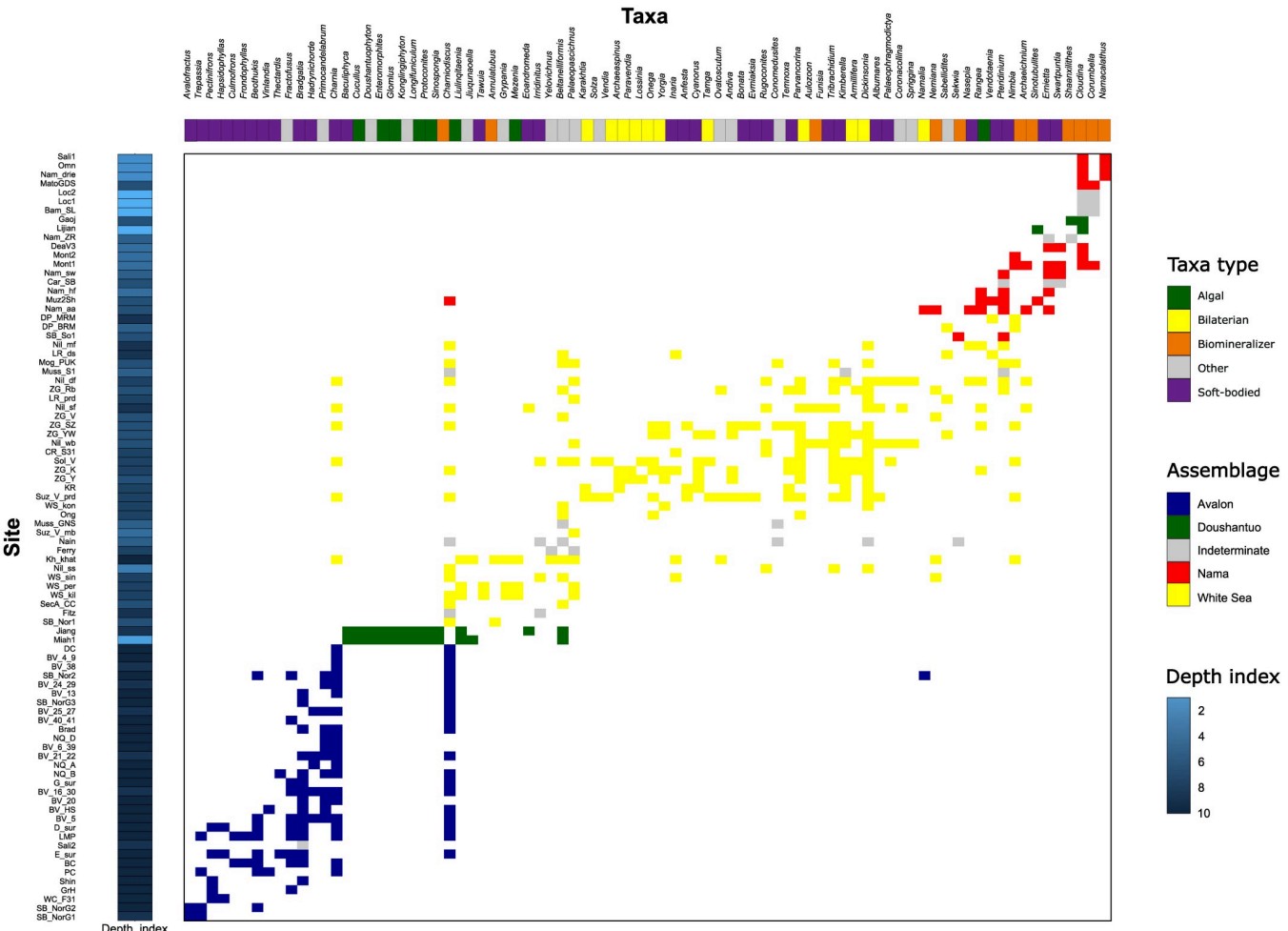

**Fig 2. Ordinated data table.** The assemblage and palaeoenvironment for each locality are given on the left. Sites are ranked based on reciprocal averaging ordination. Right plot shows the incidence matrix for taxa (columns) for all sites (rows) along the inferred environmental gradient after ordination. Ordination was calculated according to occurrence resulting from the overall metacommunity analysis. The presence of a taxon is given by a coloured square, absence by white. Doushantuo and indeterminate sites were excluded from the assemblage-level analyses. Depth index indicates the relative depth of the locality (cf., [8]), as determined by palaeoenvironment. The data underlying this Figure can be found in 10.6084/m9.figshare.13664105"https://doi.org/10.5061/dryad.1mh30 and in S1 Data.

with these assemblages, with the Avalon sites deeper, then increased shallowing up the y-axis and ordination with the Nama sites being the most shallow. This first-axis ranking of the sites was used to test whether there was a significant association with depth.

Thirdly, we will test to determine which pairwise taxa co-occurrences are significantly non-random, and whether any nonrandom co-occurrences are positive or negative. We use a combinatorics approach to test whether species pairs are randomly distributed among sites [43]. If co-occurrences are significantly nonrandom, this suggests a shared underlying ecological or evolutionary process. While the interpretation of co-occurrence data is complicated because co-occurrence does not necessarily correspond to interaction [44], here we interpret pairwise correlations (or associations) to within the wider EMS framework where co-occurrences are not taken necessarily as direct taxa interactions but could also indicate taxa environmental associations and/or disassociations.

Based on the literature, we can make predictions about how we may expect metacommunity structure to change throughout these Ediacaran assemblages. We predict that the increase

in taxonomic and morphological diversity between the Avalon and White Sea assemblages [5] is reflected in more ecological complexity in terms of increased taxa co-occurrences. We predict that the total set of Ediacaran data exhibits strong metacommunity structure that reflects the previously recovered assemblages [7–9] and that the influence of environmental gradients increases between the Avalon and White Sea assemblages [18]. Finally, we will use these analyses to test between 3 hypotheses relating to the underlying causes behind the White Sea–Nama drop in taxonomic diversity: (1) Null hypothesis: Changes in taxonomic diversity are not present or are not detectable; (2) External mass extinction: test whether there is evidence of a catastrophic extinction event between the White Sea and Nama [9,22,46]. Such an event would lead to negative turnover, so a nested metacommunity structure ([40]; or 3) Internal restructuring: increased ecological complexity via co-occurrences and strong metacommunity leading to stronger niche partitioning.

## Results

### Total Ediacaran dataset

First, we analysed all the presence/absence data of organisms as a function of sites irrespective of their assemblage, in order to test whether the assemblage definitions represented distinct communities. The sites were ranked using reciprocal averaging ordination (y-axis of Fig 2), which provided a ranking that was consistent with previous work that grouped communities into the Avalon, White Sea, and Nama assemblages [7,8]. The coherence, turnover, and boundary clumping values were calculated, and the simulated mean was used to determine if that score was significantly different (Table 1). Table 1 gives the score, simulated mean, and significant level for each of the total set, Avalon, White Sea, and Nama assemblages, and environmental subsets of the assemblages. When analysing the total dataset, we found positive coherence, turnover, and boundary clumping, characteristic of a Clementsian-structured metacommunity (Fig 3A, Table 1). Site ordination scores were significantly associated with assemblage ($H = 57.686$, $df = 2$, $p < 0.001$, Appendix Table 4), indicating strong compositional

**Table 1. Metacommunity analyses.** Metacommunity values for coherence, turnover, and boundary clumping with interpretation of metacommunity structure within the EMS framework. Z is the Z-score, $p$ is the $p$-value, and simMean is the simulated mean value of the metric.

| Group | Coherence | | | Turnover | | | Boundary Clumping | | | Interpretation |
|---|---|---|---|---|---|---|---|---|---|---|
| | Coherence | $p$ | simMean | Turnover | $p$ | simMean | Moritsita's Index | $p$ | | |
| Total | 2,230 | <0.001 | 6,690 | + | 1,270,000 | <0.001 | 1,010,000 | + | 6.390 | <<0.001 | + | Clementsian |
| Avalon | 173 | <0.001 | 309 | + | 5,120 | 0.007 | 7,650 | - | 1.880 | <0.001 | + | Clumped species loss (nested subsets) |
| Avalon (Margin slope) | 167 | <0.001 | 221 | + | 1,890 | 0.007 | 3,580 | - | 1.980 | <0.001 | + | Clumped species loss (nested subsets) |
| Avalon (Outer shelf) | 10 | 0.510 | 11 | + | 0 | 0.119 | 21 | - | 1.330 | 0.156 | + | No significant metacommunity structure |
| White Sea | 695 | <0.001 | 1,340 | + | 61,900 | 0.098 | 48,200 | + | 3.020 | <0.001 | + | Clementsian quasi structure |
| White Sea (Deep subtidal) | 60 | <0.001 | 217 | + | 2,780 | 0.230 | 2,030 | + | 3.450 | <0.001 | + | Clementsian quasi structure |
| White Sea (Middle shelf) | 182 | <0.001 | 344 | + | 5,300 | 0.110 | 3,580 | + | 4.700 | <0.001 | + | Clementsian quasi structure |
| White Sea (Outer shelf) | 0 | 0.004 | 16 | + | 66 | 0.044 | 29 | + | <0.001 | 0.227 | - | Gleasonian |
| Nama | 21 | <0.001 | 133 | + | 688 | 0.435 | 640 | + | 2.350 | <0.001 | + | Clementsian quasi structure |

EMS, Elements of Metacommunity Structure.

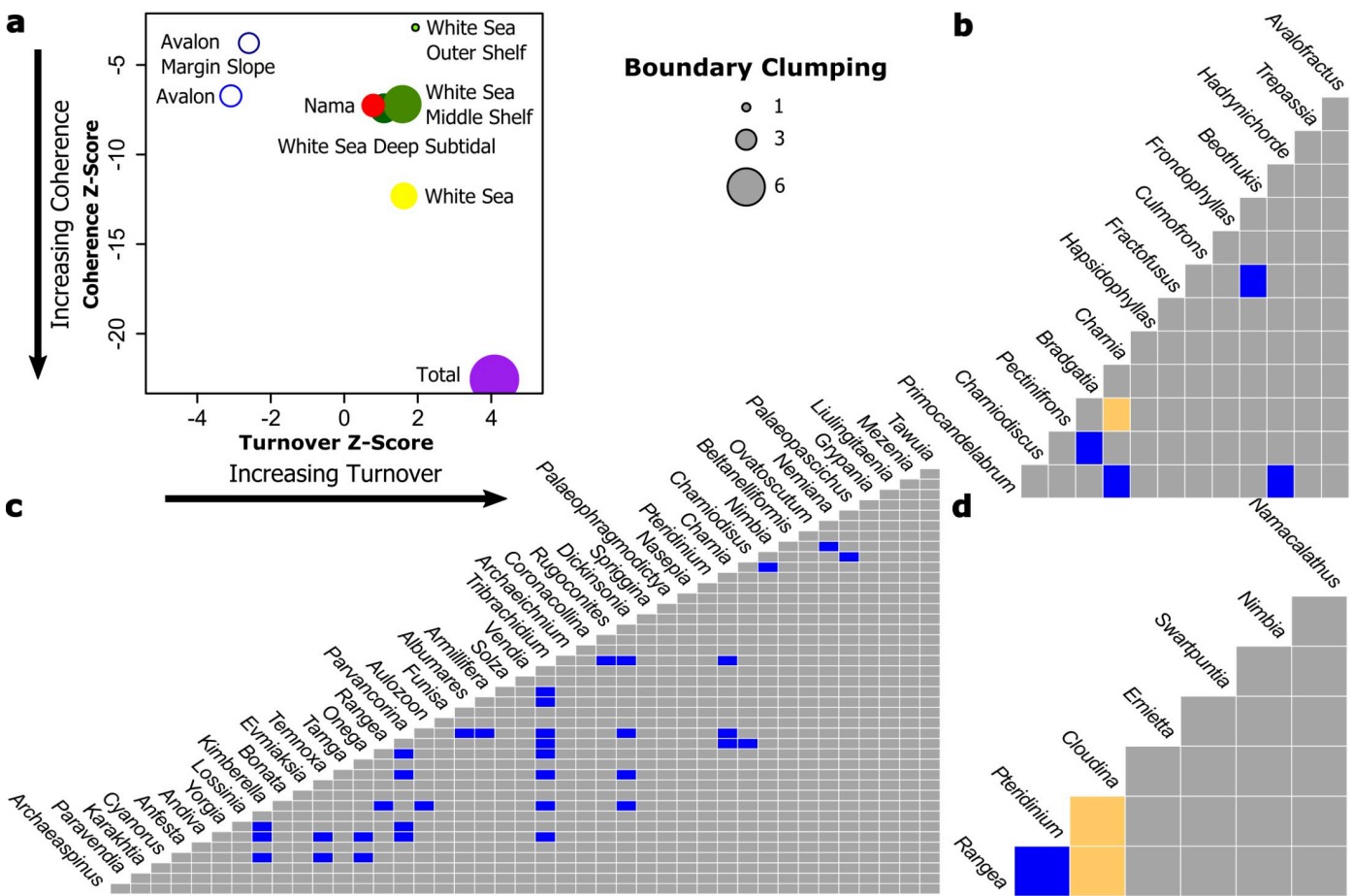

**Fig 3. Metacommunity analyses and co-occurrence matrices for each assemblage. (a)** Metacommunity plot shows a summary of the metacommunity analyses. The z-score is a standardised way to indicate how much the observed means differ from the average across all communities in terms of standard deviations. Nested species loss is shown by an open circle; Clementsian by a closed circle; Gleasonian by a black ringed circle. The size of the circle represents the value of the boundary clumping score. The Avalonian outer shelf had a random structure and so is not shown. Co-occurrence matrices for **(b)** the Avalon metacommunity, **(c)** the White Sea metacommunity, and **(d)** the Nama metacommunities. Positive associations are blue; negative associations are yellow.

difference among the assemblages. Site scores were also significantly correlated with depth ($\rho = -0.427$, $p < 0.001$, Table 2), suggesting that the structuring in the dataset may be due to the differences in depth between sites. However, since depth also significantly varies with assemblage ($H = 53.987$, $df = 2$, $p < 0.001$, Table 2), it is not possible to tell whether the structuring is due to depth or another factor that varies with assemblage.

**Table 2. Reciprocal averaging analyses.** Spearman rank correlations ($R_s$) (for continuous variables) and Kruskal–Wallis tests (for discrete variables) between site scores of each dataset obtained from reciprocal averaging and a variable, either assemblage or depth.

| | $R_s$ | Kruskal–Wallis | df | *p*-value |
|---|---|---|---|---|
| All assemblages site scores × assemblage | - | 57.686 | 2 | <0.001 |
| All assemblages site scores × depth | −0.427 | - | - | <0.001 |
| Depth × assemblage | - | 53.987 | 2 | <0.001 |
| Avalon site scores × depth | −0.360 | - | - | 0.051 |
| White Sea site scores × depth | 0.014 | - | - | 0.945 |
| Nama site scores × depth | −0.728 | - | - | 0.007 |

To further investigate the nature of this structuring, we focused on the pairwise co-occurrence patterns, finding 10.3% were nonrandom: All but one positive associations resulted from taxa specific to the same assemblage (96.8% of positive associations), and all negative associations from taxa exclusively found in or heavily more abundant in different assemblages to each another (S2 Table) (100% of negative associations). The one exception was a positive association between *Pteridinium* and *Rangea*: These taxa are found roughly equally in Nama and White Sea sites with only a slight skew towards one or the other (S5 and S7 Tables). Therefore, our analyses are consistent with previous studies [8,9] in finding that Ediacaran taxa are highly segregated by assemblage, as well as confirming a role for depth (Table 2) in structuring the assemblages [7,8]. At this broad level of analysis, the strong assemblage signal, at least partially dependent on depth specialisations, obscures any other biotic or abiotic pattern.

## Avalon metacommunity analyses

We then narrowed our level of analysis by focusing on each assemblage in turn. The Avalonian metacommunity displays significant positive coherence and boundary clumping, but significant negative turnover (Fig 3, Table 1), characteristic of a pattern of "nested clumped species loss" [37,45,46]. Site ordination scores were not significantly correlated with depth ($Rs = -0.360$, $p = 0.051$, Table 2) and so depth is only weakly associated with this metacommunity structuring. For Avalonian taxa, segregation was rare (Fig 3): Only one negative taxa association was found— between *Charnia* and *Pectinifrons* (Fig 3, S3 Table). Positive associations were found between *Bradgatia* and *Charniodiscus*, *Fractofusus* and *Beothukis*, *Primocandelabrum*, *Charnia*, *Hadrynichorde*, and *Primocandelabrum* (Fig 3, Table 2). The metacommunity structure was the same for the Avalonian margin slope metacommunity (the associations between *Bradgatia* and *Charniodiscus* and *Fractofusus* and *Beothukis* remained), but the outer shelf showed no significant structuring (Fig 2, S3 Fig, Table 1, S3 and S4 Tables). The predominance of positive over negative taxa co-occurrences is consistent with previous studies of detailed within-community spatial analyses of focal taxa, which found little evidence for lateral resource competition between Avalonian taxa [11,18]. The lack of depth and palaeoenvironmental correlation with metacommunity structure supports suggestions that Avalonian organisms have the widest niches and lowest provinciality among the Ediacaran biotas [7,12].

The Avalonian metacommunity displays a structure of "nested clumped species loss," whereby taxa-poor communities form nested subsets of increasingly taxa-rich communities, with predictable patterns of taxa loss associated with variation in taxa characteristics [30]. Differences in Avalonian community composition have been suggested to represent different stages of community succession, based on community parameters, cluster analyses and MDS (Multidimensional Scaling) ordination [13]. Where multiple different stages of a community succession are analysed using EMS, the succession would result in the observed pattern of clumped taxa loss with early and late succession communities forming less diverse nested subsets of maximally diverse mid-succession communities. This metacommunity structure and proposed succession is consistent with Connell's disturbance theory [13] whereby intermediate stages of a community are most diverse because they enable both early and late colonisers to coexist [47]. However, the lack of interspecific competition found by previous studies using spatial analyses within communities of early-stage communities [11,12,14] suggests that interactions other than competitive exclusion were influencing community development. Avalonian metacommunity structure had not previously been statistically compared to multiple different models (here 14 different models). Our results find that the Avalonian metacommunity exhibits clumped taxa loss, which supports Connell's model. This model is further supported by the co-occurrence analyses, which found a negative association between late-

succession and early-stage taxa (*Charnia* and *Pectinifrons*), and positive associations between late-stage (*Primocandelabrum* and *Charnia*), middle-stage (*Bradgatia* and *Charniodiscus*), and early-stage (*Beothukis* and *Fractofusus*) stage taxa (S2 Fig, S4 Table) [13].

The taxa associations of the Avalonian sites in the UK form a subset of the associations from the Canadian sites, suggesting structuring is not due to geographic or abiotic differences between the 2 areas (S3 Fig, S4 Table). Upon removing the UK sites from the analysis, positive associations between *Fractofusus* and *Beothukis*, and *Hadrynichorde* and *Primocandelabrum* remained significant (extended data Fig 4, S3 and S4 Tables). The association between *Charniodiscus* and *Bradgatia* was still present although not significant ($p = 0.134$) and the association between *Charnia* and *Primocandelabrum* was weakly significant ($p = 0.054$). The negative association between *Charnia* and *Pectinifrons* was also nearly significant ($p = 0.089$). There were also no new significant positive associations when studying only the Canadian sites and the associations thus form a perfect subset of the total dataset (S4 Fig, S6 Table). In Mistaken Point, *Charnia* is dominant in proposed late succession communities (on Lower Mistaken Point), whereas *Pectinifrons* is characteristic of early succession communities (on Shingle Head and D surface) [10,13]. These 2 taxa show a negative co-occurrence. *Primocandelabrum*, like *Charnia*, is characteristic of late- and middle-stage succession communities. *Bradgatia* and *Charniodiscus* are both overwhelmingly present in mid-succession communities. *Beothukis* and *Fractofusus* are most heavily found in early-mid-succession communities [13]. These pairs of taxa demonstrate positive co-occurrences. The Charnwood sites notably lack *Fractofusus* and *Pectinifrons* [48], both very characteristic of proposed early succession sites in Canada, but share many of the same taxa that are seen in proposed middle- and late-stage succession sites in Canada (*Charnia*, *Primocandelabrum*, *Bradgatia*, and *Charniodiscus*). There have been many fewer UK sites sampled than in Newfoundland, so it is plausible that the current communities do not reflect the full diversity of the area [49]. Thus, our results provide evidence that the UK and Canadian sites represent communities along similar community successions. Communities in the Flinders Range, South Australia (part of the White Sea assemblage) have been proposed to show evidence of primary succession [50], and so succession may be characteristic of many Ediacaran, as well as modern, communities. Therefore, observed differences in community composition between UK and Canadian Avalonian sites may reflect which stages in community development are preserved, and thus represent a biotic rather than a geographic signal.

### White Sea metacommunity analyses

The White Sea metacommunity displays significant positive coherence and boundary clumping and nonsignificant positive turnover, characteristic of a "Clementsian quasi structure" (S1 Table) [30] with no correlation between site ordination scores and depth ($Rs = 0.014$, $p = 0.945$, Fig 2; Tables 1 and 2), and so depth alone is unlikely to be responsible for this metacommunity structure (Table 2). Metacommunity structure within the White Sea assemblage differed with palaeoenvironment: The deep subtidal and middle shelf metacommunities displayed "Clementsian quasi structures," while the outer shelf metacommunity was characterised by a Gleasonian structure, with significant positive coherence and turnover and nonsignificant boundary clumping (Fig 3, S4 Fig, Table 1). Co-occurrence analyses found that 11 of the 32 positive associations found for the whole assemblage were preserved when focusing on the palaeoenvironmental subsets (Fig 3, S4 Fig, S6 and S7 Tables). A positive association between *Parvancorina* and *Tribrachidium* was the only association to appear in both environmental subdivisions despite almost all the White Sea taxa being present in both the middle shelf and deep subtidal environments. In deep subtidal facies, there is a notable negative association between *Beltanelliformis* and *Kimberella* (S5 Fig, S5–S7 Tables).

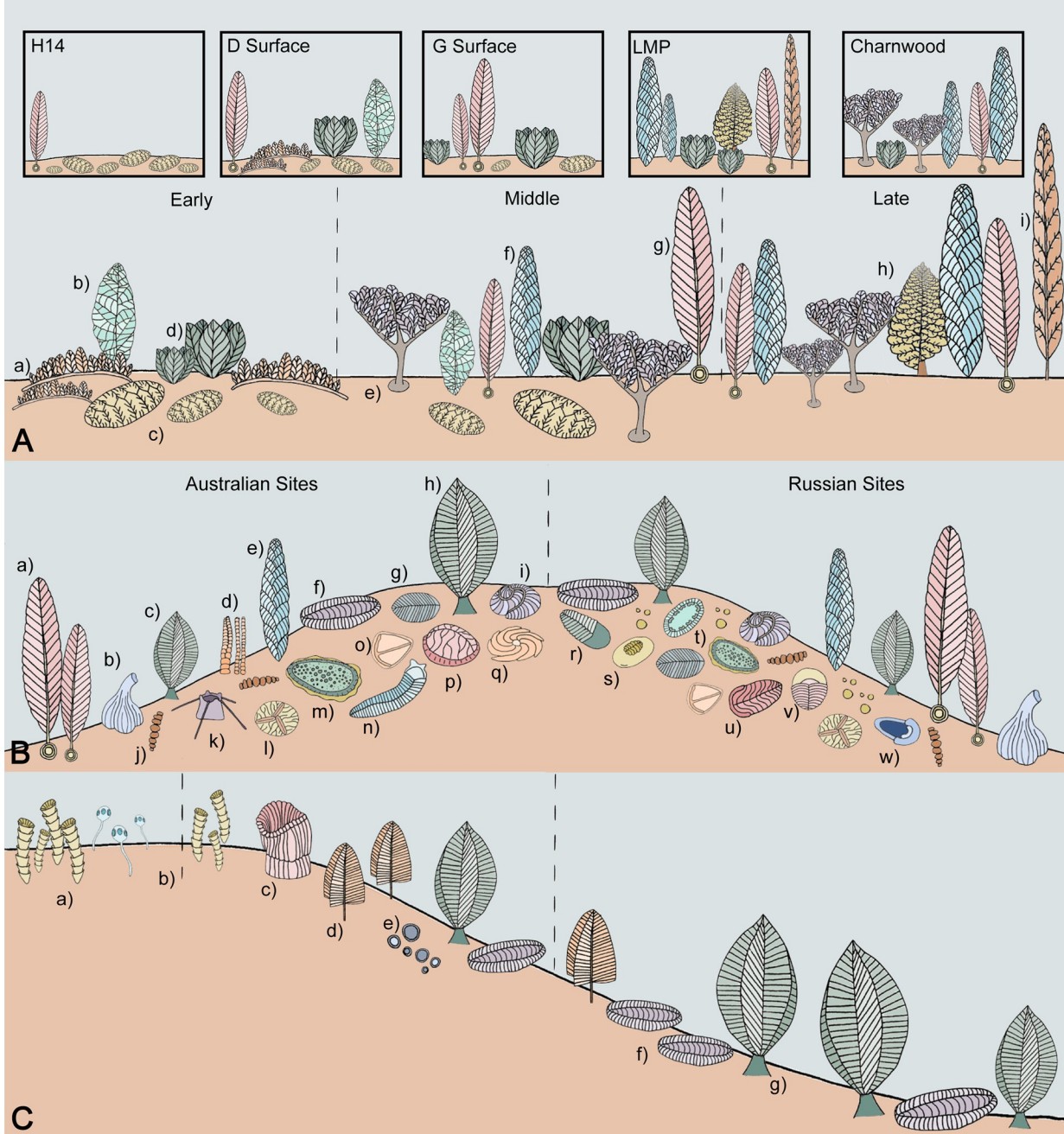

**Fig 4. Reconstructions of the Avalon, White Sea, and Nama metacommunities. (A)** A reconstruction of the Avalon assemblage showing the proposed stages of a community succession with the actual composition of several surfaces in boxes above. (a) *Pectinifrons*; (b) *Beothukis*; (c) *Fractofusus*; (d) *Bradgatia*; (e) *Primocandelabrum*; (f) *Charnia*; (g) *Charniodiscus*; (h) *Culmofrons*; (i) *Trepassia*. **(B)** A reconstruction of the White Sea assemblage showing some endemism of taxa to the Russian or Australian sites. (a) *Charniodiscus*; (b) *Inaria*; (c) *Rangea*; (d) *Funisia*; (e) *Charnia*; (f) *Pteridinium*; (g) *Dickinsonia*; (h) *Rangea*; (i) *Tribrachidium*; (j) *Palaeopaschinus*; (k) *Coronacollina*; (l) *Albumares*; (m) *Kimberella*; (n) *Spriggina*; (o) *Parvancorina*; (p) *Rugoconites*; (q) *Eoandromeda*; (r) *Cyanorus*; (s) *Onega*; (t) *Armillifera*; (u) *Andiva*; (v) *Yorgia*; (w) *Temnoxa*. **(C)** A reconstruction of the Nama assemblage showing the palaeoenvironmental separation of biomineralising and soft-bodied taxa across a depth profile. (a) *Cloudina*; (b) *Namacalthus*; (c) *Ernietta*; (d) *Swartpuntia*; (e) *Nimbia*; (f) *Pteridinium*; (g) *Rangea*. Taxa and environmental separation are not to scale. LMP, Lower Mistaken Point surface.

Several taxa seemed to have unique taxa associations in each subdivision despite very similar community composition. (S4 Fig, S6 and S7 Tables), which naively we would expect to lead to the same taxa associations. The 11/32 positive associations that differed between the middle shelf and deep subtidal environments show that the community associations are nonconsistent between the subsets and the assemblage as a whole. The underlying processes that contribute to these differences in both intertaxa interactions and environmental factors could be due to organism behavioural plasticity, leading to different behaviours in different environments. Alternatively, differences in taxa associations for a given taxon may reflect the inclusion of several taxa with different environmental preferences and behaviours within one taxonomic group (e.g., *Dickinsonia*). However, as most Ediacaran taxonomic groupings are monotypic the differences in taxa associations in different environments are more likely due to plastic responses to, e.g., variation in resource limitation or the presence of different competitors or ecosystem engineers.

The majority of taxa pairwise associations found in the Russian White Sea are also found in the pooled White Sea metacommunity (24/37). However, 13 of the pooled associations are present when only analysing Russian localities, suggesting that some of the structure in the dataset may be due to geography (S5 and S8 Tables). There is marked geographic variation in community composition between the Russian and Australian White Sea localities [24], so the nonshared associations may reflect a greater endemism within the White Sea assemblage compared to the Avalonian assemblage, where the UK sites formed a perfect subset of the Canadian sites.

The only evidence of a putative consumer–resource interaction was found in the White Sea metacommunity, which is consistent with the idea that grazing and motility evolved as part of the "second-wave radiation," where the first-wave radiation was the "Avalon Explosion" [5,51,52]. In White Sea deep subtidal facies, there was a negative association between *Beltanelliformis* and *Kimberella*. This result may be a consequence of herbivory as *Kimberella* has been reconstructed as a mobile grazer [53,54] and *Beltanelliformis* as large colonies of cyanobacteria [55]. It has also previously been noted that remains of *B. brunsae* sometimes co-occur with the feeding traces of *Kimberella* (*Kimberichnus teruzzii*) [56,57].

## Nama metacommunity analyses

The Nama metacommunity has significant positive coherence and boundary clumping and nonsignificant positive turnover and so displays a "Clementsian quasi structure," the same metacommunity structure as the White Sea assemblage (Table 1) [30]. Unlike the Avalon and White Sea sites, ordination scores were significantly correlated with depth ($Rs = -0.728$, $p = 0.007$, Table 2), and so the Clementsian structuring occurs along a depth gradient. Pairwise taxa co-occurrences revealed significant negative associations between biomineralisers (*Cloudina*) and soft-bodied taxa (*Pteridinium* and *Rangea*) and a significant positive association between 2 soft-bodied taxa (Fig 3, S10 Table). There were more negative than positive associations (Fig 3, S10 Table). These results statistically confirm previous observations of separation between biomineralisers (e.g., *Cloudina*) and soft-bodied organisms (such as *Pteridinium* and *Rangea*) [58].

These patterns of segregation are unlikely to be purely facies-based control for our data set, which is purely a result of the chemical and physical properties of the rock that the fossils are preserved in. In the Nama assemblage, there are both deep and shallow water carbonate facies. The deep water Nama facies (Dengying Fm) has a more similar community composition to the other deep water Nama sites than to the carbonate sites and thus help to strengthen the pattern of taxa segregation by depth as opposed to counteract it (which is what we would expect if

there was purely facies-based control of taxa separation). If the habitat specialisation was a reflection on biomineralisation alone, that we would expect to see the biomineralisers behave in broadly similar patterns, and the soft-bodied taxa to also behave similarly to each other. Of the 7 taxa that were sufficiently abundant to be included in these analyses, one was a putative microbial colony (*Nimbia* [59]), 2 were biomineralisers (*Cloudina* and *Namacalathus*), and the remaining 4 were soft-bodied taxa (*Rangea*, *Pteridinium*, *Ernietta*, and *Swartpuntia*). *Nimbia* did not show any significant associations with the other taxa, although (like *Namacalathus*) it was only present in 2 sites, this lack of associations may be due in part to small sample sizes. The 2 biomineralisers behaved in different ways—while *Cloudina* showed negative associations with the soft-bodied *Rangea* and *Pteridium*. *Namacalathus* did not show any significant associations with any taxa but could be biased by the number of sites in which it is present. The soft-bodied taxa also did not behave in a uniform way, with *Ernietta* and *Swartpuntia* displaying no significant associations with either other soft-bodied taxa nor biomineralisers while *Rangea* and *Pteridium* showed significant positive association with each other and negative associations with *Cloudina*. As such, there are no consistent patterns of biomineralisers nor soft-bodied taxa that explain the patterns found within our data.

As such, the signal in our data cannot be attributed solely to carbonate/siliciclastic nor biomineralisers/soft-bodied taxa differences, and so is most likely due to habitat preferences as community composition was found to vary significantly with depth. *Cloudina* is found exclusively in shallow limestone and shallow siliciclastic shoreface facies, whereas soft-bodied Nama taxa are found in both deeper shoreface and deep subtidal settings (Fig 4).

## Effect of sampling biases

The Nama assemblage has notably less localities [9] than either the Avalon [29] or the White Sea [28], which could suggest that differences in the Nama are merely an artefact of sampling. Therefore, it is important to understand how these sampling differences could affect the EMS analyses and the co-occurrence analyses. We assessed these biases in 2 ways: (1) by comparing results of environmental subsets of the Avalon and White Sea, which are similar in size to the Nama assemblage; and (2) by simulating Avalon and White Sea data by subsampling the larger datasets to that of Nama—9 localities then testing for significant nonrandom co-occurrences and for a correlation between site score and depth.

First, in terms of co-occurrence, for the Avalon subsets, the margin slope [23] and outer shelf [6] have 4.5% and 0% significant nonrandom co-occurrence, and for the White Sea, the deep subtidal has 4.1% significant co-occurrences and middle shelf has 7.9%. These values are much smaller than that of the Nama at 16.7%. Furthermore, they show an increase from the Avalon to the White Sea, thus confirming the overarching pattern of increasing co-occurrences found in the full sets. In terms of metacommunity structure, the Avalon environmental subsets have the same metacommunity structure (nested clumped species loss) as the whole Avalon, with negative turnover and small coherence factors. Similarly, the White Sea environmental subsets both have the same quasi-Clementsian structure as the whole White Sea assemblage. Thus, the changes of metacommunity structure from the Avalon to the White Sea are maintained within the environmental subsets with sample sizes similar to those available for Nama.

Second, we performed randomised tests for the co-occurrence and depth analyses for the Avalon and White Sea data, subsampling the datasets 1,000 times each from 29 (Avalon) and 28 (White Sea) to 9 (Nama). Avalon had significantly less nonrandom co-occurrences than the While Sea and Nama ($p_{av.vs.ws} = 0.016$; $p_{av.vs.nm} = 0.016$), in contrast to the White Sea, which showed no significant difference in numbers of co-occurrences ($p_{ws.vs.nm} = 0.158$). In order to

test for the significance of the depth correlation we performed Spearman test each of the 1,000 subsampled data. Only a small number of subsamples showed a significant correlation for depth (24 out 1,000 for Avalon; and 17 out of 1,000 for the White Sea). Therefore, we are confident that our results showing an increase in co-occurrence between the Avalon and Nama and an increase in depth structure between the White Sea and Nama are not artefacts of different sampling, but robust signals.

## Discussion

The total set of Ediacaran data exhibits strong metacommunity structure, consistent with previous analyses that resolve multiple assemblages [7–9]. Both the total dataset and the individual assemblages have relatively low numbers of nonrandom co-occurrences (9.8% to 16.7%) compared to many extant analyses (e.g., 35% to 63% [60,61]) as well as terrestrial fossil communities (such as averaging 64% aggregated pairs from the Carboniferous to the Holocene, and 37% from the Holocene to the present [62], although percentages of nonrandom co-occurrences are similar to at least some extant benthic communities (16.3%) [63]. Previous spatial analyses of Avalonian communities have revealed limited interspecific interactions [11,14] and limited environmental associations between taxa within communities [12], so the large number of nonsignificant correlations within the Avalon assemblage are consistent with previous work. Given that the oldest assemblage is dominated by non-co-occurrences, it follows that the subsequent development of Ediacaran metacommunities would have to build from this point and so are not immediately comparable to extant communities with longer evolutionary histories. There is strong evidence for the second-wave diversification reflected in our results in an increase of nonrandom taxa associations from the Avalonian biota (9.8%) to the White Sea biota (16.1%, $p_{avalon}$ = 0.016). Further evidence for this increased ecological complexity (i.e., greater interactions and associations between taxa) is provided by the fact that the Avalonian assemblage had minimal environmental structuring, while we detected both Gleasonian and quasi-Clementsian metacommunity structure depending on the palaeoenvironments for the White Sea assemblage. While there is not a significant correlation of broad-scale palaeoenvironment with metacommunity structure, these structures reflect a significant influence of a fine-scale environmental gradient. Gleasonian structuring reflects an individualistic response to the inferred environmental gradient, suggesting a lack of within-community associations for this outer shelf metacommunity. In contrast, quasi-Clementsian structure corresponds to a community-wide response to the environmental gradients, so reflects within-community specialisations in the White Sea assemblage. These specialisations are reflected in behavioural flexibility, with organisms exhibiting different taxa associations in different environments across a wide range of depths.

The White Sea and Nama assemblages have the same type of metacommunity structure as shown through EMS, with both assemblages showing a quasi-Clementsian structure (Fig 3, Table 1). There is a small increase in nonrandom associations in the Nama biota (16.7% from 16.1% in the White Sea), which, like the EMS analyses, shows at least a maintenance, if not slight increase, in metacommunity structure. There is a significant increase in ecosystem structuring between the White Sea and Nama assemblages when site rank within each of the assemblages are compared to depth (Table 2). Neither the Avalon nor White Sea show such significant correlation with depth, in sharp contrast to the Nama, where site composition is significantly correlated with depth (Table 2). Taken together, these 3 analyses show that compared to the White Sea, the Nama assemblage has an increased taxa segregation coupled to a strong palaeoenvironmental correlation and therefore narrowed environmental tolerances, showing a further decrease of niche breadth. Thus, we have shown that the increase in

complexity of the taxon-specific ecological strategies utilised throughout the Ediacaran is mirrored in the complexity of the community associations.

A White Sea–Nama catastrophic environmental extinction is in consistent with our results for 3 reasons. Firstly, a catastrophic mass extinction implies that surviving taxa within the Nama assemblage are more likely to be generalists [8,9,64], contrary to our results. We have shown an increased influence of paleoenvironment and niche specialisation with the Nama metacommunity showing significant correlation between community composition and depth, in contrast to the White Sea and Avalon metacommunities. Second, the Nama metacommunity exhibits a nonsignificant but positive turnover, indicating more turnover along the gradient (more niche differentiation). If the Nama assemblage metacommunity structure was due to underlying extinction/colonisation dynamics, we expect to see an increase in nestedness [40], as indicated by negative turnover, contrary to our positive Nama turnover (Table 1). Thirdly, this increase in turnover suggests not only higher ecosystem complexity but also increased taxa specialisation and narrower niche breadth coupled with an increase in within-community structuring between the White Sea and Nama assemblages, with a slight increase in nonrandom taxa associations (16.1% to 16.7%). A decrease in taxonomic and morphological diversity in the Nama [23] may reflect that, within this assemblage, Ediacaran organisms show significant palaeoenvironmental preferences, and thus reduced environmental tolerances, resulting in multiple different types of mutually exclusive communities, each of which exhibits a simple structure within its narrow niche [23]. An increase in within-community structure in the form of ecosystem engineering [65] and reef complexity [20] provides supporting evidence that despite a decrease in taxonomic diversity, the Nama assemblage represents an ecological development from the White Sea assemblage, not a recovery from a catastrophic extinction event. Our results are further supported by birth–death models of stem and crown group diversification, which predict Ediacaran-like diversification patterns for bilaterians and produce patterns that can be easily mistaken for mass extinctions [66].

Our results show that these Ediacaran organisms underwent the niche contraction and specialisation that is traditionally associated with Cambrian diversification [6,67]. Therefore, we find that the eco-evolutionary dynamics of metazoan diversification known from the Cambrian started earlier in the Ediacaran with the Avalon assemblage and increased in complexity towards the Phanerozoic as new anatomical innovations appeared, culminating in the "Cambrian Explosion."

## Materials and methods

### Materials

The data used in this study is a binary presence/absence matrix for 86 Ediacaran localities and 124 taxa. The data is taken from [9] with more conservative classifications of assemblages for several sites (cf., [8]): SB-Nor2 and SB-So1, both Sewki Brook sites, are classed as indeterminate in our analysis and [8] but as Avalon and Nama, respectively, in [9]. Two Chinese sites classed as Nama by Muscente and colleagues are classed as indeterminate here (Gaojiashan and Lijiagou) [8,24]. The data contains information on the palaeoenvironment, depth index (from 1 to 11), lithology, and assemblage of each locality as well as a time index (from 1 to 3). The full classification of sites in the dataset can be seen in S1 Data and the palaeoenvironmental and assemblage classifications in Fig 1. This data is appropriate for applying modern statistical ecological methods because the organisms were mostly sessile and benthic and preserved in such a way that they are interpreted as in situ life assemblages with minimal transportation after death or time-averaging [13,26–28].

The Avalon, White Sea, and Nama assemblages are represented by 30, 29, and 12 sites, respectively, with 11 undetermined sites and 4 that belong to the Doushantuo assemblage. The Doushantuo assemblage was excluded from the assemblage-level analyses because there are only 4 sites, which is insufficient to run these analyses.

Secondly, we used Spearman rank correlations to test whether within-assemblage community composition is correlated with depth. The ordering of the sites was given by the ordination output from the EMS analyses (Fig 2), which is produced by reciprocal averaging, a type of correspondence analysis that ordinates the sites based on their species composition [31]. This ordering provides the first-axis ranking of the sites that were then used to test whether there was a significant association with depth.

## Methods

The R package Metacom was used for the EMS analyses [37]. The first step of the EMS analyses was to use reciprocal averaging to ordinate the sites based on their species composition ([31]; Fig 2). Metacommunity structure is then quantified based on this ordering via the calculation of the 3 metrics related to metacommunity structure: coherence, turnover, and boundary clumping [37]. A 3-tiered analysis based on these metrics enabled the placement of each studied metacommunity into one of 14 idealised metacommunity structures following the EMS approach [29,30]. In the context of EMS, statistical significance is calculated using a z-test (i.e., calculating the z-score) of the observed absences to embedded absences in randomised null matrices [31]. For each subsection of data (e.g., each assemblage), the z-score is calculated relative to that subset only, rather than the total set of data. Significant coherence is a prerequisite for further analysis of metacommunity structure, and so this was the first metric calculated. Calculation of turnover was then used to distinguish whether the metacommunity formed a nested structure. Calculation of boundary clumping allowed the determination of whether the taxa range boundaries were clumped, dispersed, or not significantly correlated with each other along the environmental gradient. Fig 1 gives examples of taxa abundance distributions that give rise to the idealised metacommunity structures. Sites scores were given according to the ranking of the sites in the first-degree ordination of their taxa composition. These were used to investigate the importance of depth (as indicated by palaeoenvironment) and assemblage variables in structuring taxa distributions via a nonparametric Spearman correlation or Kruskal–Wallis test. The metacommunity analyses were performed on the entire dataset, for each assemblage individually, and for palaeoenvironmental and geographic subsets within each assemblage where there were enough localities for valid analyses.

## Co-occurrence analysis

The R package co-occur was used to calculate the observed and expected frequency of co-occurrence between pairs of taxa to determine significant positive or negative associations [42]. Taxa, which only occurred in 1 site, were removed from the analysis because such singletons have been shown to disproportionally influence co-occurrence analyses [68,69]. Co-occurrence analysis was done for the entire dataset, for each assemblage individually, and for palaeoenvironmental and geographic subsets within each assemblage where there were enough sites [42].

## Supporting information

**S1 Fig. Locality map showing Ediacaran sites.** The names, assemblages, and various characteristics of each of the localities can be found in Fig 1 and in S1 Data. The map is from

generated in R using the ggplot2 and sf packages in R, using an OpenStreetMap basemap (71).
(EPS)

**S2 Fig. Co-occurrence matrix for the total dataset.** Co-occurrence matrix for species showing significant associations in the whole dataset. Positive associations are blue; negative associations are yellow.
(EPS)

**S3 Fig. Co-occurrence matrices for the Avalonian dataset and subsets.** Co-occurrence matrices for species showing significant associations in (**a**) the Canadian Avalonian metacommunity, (**b**) the Avalonian margin slope metacommunity, and (**c**) the Avalonian outer shelf metacommunity. Positive associations are blue; negative associations are yellow. The Avalonian outer slope metacommunity had no significant associations.
(EPS)

**S4 Fig. Co-occurrence matrices for the White Sea dataset and subsets.** Co-occurrence matrices for species showing significant associations in (**a**) the Russian White Sea metacommunity, (**b**) the White Sea middle shelf metacommunity, and (**c**) the White Sea deep subtidal metacommunity. Positive associations are blue; negative associations are yellow.
(EPS)

**S1 Table. Summary table of how metacommunity properties are expressed in terms of the EMS metrics.**
(DOCX)

**S2 Table. Co-occurrence analysis for the total dataset showing only significant associations.** Sp1_inc is the number of sites that have taxa 1. Obc_cooccur is the observed number of sites with both species. Prob_cooccur is the probability both species occur at a site. Exp_cooccur is the expected number of sites having both taxa. P_Lt is the probability that the 2 taxa would co-occur at a frequency less than observed, and P_gt is the probability that the 2 taxa would co-occur at a frequency greater than observed. Difference is the difference between observed and expected probabilities, where difference > 0.95 the association is considered significant.
(DOCX)

**S3 Table. Co-occurrence analysis for the Avalonian dataset showing only significant associations.** Sp1_inc is the number of sites that have taxa 1. Obc_cooccur is the observed number of sites with both species. Prob_cooccur is the probability both species occur at a site. Exp_cooccur is the expected number of sites having both taxa. P_Lt is the probability that the 2 taxa would co-occur at a frequency less than observed, and P_gt is the probability that the 2 taxa would co-occur at a frequency greater than observed. Difference is the difference between observed and expected probabilities, where difference > 0.95 the association is considered significant.
(DOCX)

**S4 Table. Co-occurrence analysis for the Avalonian Canada dataset showing only significant associations.** Sp1_inc is the number of sites that have taxa 1. Obc_cooccur is the observed number of sites with both species. Prob_cooccur is the probability both species occur at a site. Exp_cooccur is the expected number of sites having both taxa. P_Lt is the probability that the 2 taxa would co-occur at a frequency less than observed, and P_gt is the probability that the 2 taxa would co-occur at a frequency greater than observed. Difference is the difference between observed and expected probabilities, where difference > 0.95 the association is considered

significant.
(DOCX)

**S5 Table. Co-occurrence analysis for the Avalonian margin slope dataset showing only significant associations.** Sp1_inc is the number of sites that have taxa 1. Obc_cooccur is the observed number of sites with both species. Prob_cooccur is the probability both species occur at a site. Exp_cooccur is the expected number of sites having both taxa. P_Lt is the probability that the 2 taxa would co-occur at a frequency less than observed, and P_gt is the probability that the 2 taxa would co-occur at a frequency greater than observed. Difference is the difference between observed and expected probabilities, where difference > 0.95 the association is considered significant.
(DOCX)

**S6 Table. Co-occurrence analysis for the White Sea dataset showing only significant associations.** Sp1_inc is the number of sites that have taxa 1. Obc_cooccur is the observed number of sites with both species. Prob_cooccur is the probability both species occur at a site. Exp_cooccur is the expected number of sites having both taxa. P_Lt is the probability that the 2 taxa would co-occur at a frequency less than observed, and P_gt is the probability that the 2 taxa would co-occur at a frequency greater than observed. Difference is the difference between observed and expected probabilities, where difference > 0.95 the association is considered significant.
(DOCX)

**S7 Table. Co-occurrence analysis for the White Sea middle shelf dataset showing only significant associations.** Sp1_inc is the number of sites that have taxa 1. Obc_cooccur is the observed number of sites with both species. Prob_cooccur is the probability both species occur at a site. Exp_cooccur is the expected number of sites having both taxa. P_Lt is the probability that the 2 taxa would co-occur at a frequency less than observed, and P_gt is the probability that the 2 taxa would co-occur at a frequency greater than observed. Difference is the difference between observed and expected probabilities, where difference > 0.95 the association is considered significant.
(DOCX)

**S8 Table. Co-occurrence analysis for the White Sea deep subtidal dataset showing only significant associations.** Sp1_inc is the number of sites that have taxa 1. Obc_cooccur is the observed number of sites with both species. Prob_cooccur is the probability both species occur at a site. Exp_cooccur is the expected number of sites having both taxa. P_Lt is the probability that the 2 taxa would co-occur at a frequency less than observed, and P_gt is the probability that the 2 taxa would co-occur at a frequency greater than observed. Difference is the difference between observed and expected probabilities, where difference > 0.95 the association is considered significant.
(DOCX)

**S9 Table. Co-occurrence analysis for the White Sea Russian dataset showing only significant associations.**
(DOCX)

**S10 Table. Co-occurrence analysis for the Nama dataset showing only significant associations.** Sp1_inc is the number of sites that have taxa 1. Obc_cooccur is the observed number of sites with both species. Prob_cooccur is the probability both species occur at a site. Exp_cooccur is the expected number of sites having both taxa. P_Lt is the probability that the 2 taxa would co-occur at a frequency less than observed, and P_gt is the probability that the 2 taxa

would co-occur at a frequency greater than observed. Difference is the difference between observed and expected probabilities, where difference > 0.95 the association is considered significant.
(DOCX)

**S1 Data. Supplementary data.**
(CSV)

**S1 Code. Supplementary code.**
(TXT)

## Author Contributions

**Conceptualization:** Andrea Manica, Emily G. Mitchell.

**Data curation:** Rebecca Eden, Emily G. Mitchell.

**Formal analysis:** Rebecca Eden, Emily G. Mitchell.

**Funding acquisition:** Emily G. Mitchell.

**Methodology:** Andrea Manica, Emily G. Mitchell.

**Project administration:** Emily G. Mitchell.

**Supervision:** Andrea Manica, Emily G. Mitchell.

**Visualization:** Rebecca Eden, Emily G. Mitchell.

**Writing – original draft:** Rebecca Eden, Andrea Manica, Emily G. Mitchell.

**Writing – review & editing:** Andrea Manica, Emily G. Mitchell.

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
