## [Editor Report · Decision Letter 0]

12 May 2021

Dear Dr Mitchell, 

Thank you for submitting your manuscript entitled "Metacommunity analyses show increase in ecological specialisation throughout the Ediacaran" for consideration as a Research Article by PLOS Biology.

Your manuscript has now been evaluated by the PLOS Biology editorial staff, as well as by an academic editor with relevant expertise, and I'm writing to let you know that we would like to send your submission out for external peer review.

Please re-submit your manuscript within two working days, i.e. by May 14 2021 11:59PM.

Kind regards,

Roli Roberts

Roland Roberts

Senior Editor

PLOS Biology

rroberts@plos.org

---

## [Decision Letter · Decision Letter 1]

16 Jul 2021

Dear Dr Mitchell,

Thank you very much for submitting your manuscript "Metacommunity analyses show increase in ecological specialisation throughout the Ediacaran" for consideration as a Research Article at PLOS Biology. Your manuscript has been evaluated by the PLOS Biology editors, an Academic Editor with relevant expertise, and by three independent reviewers. Please accept my apologies for the delay in delivering a decision; we've had the reviews for a while but had difficulties contacting the Academic Editor to discuss the way forward.

IMPORTANT: You'll see that the reviewers are all broadly positive about your study. However, they clearly feel that the manuscript needs a lot of work in two main areas: first improving the clarity, flow and structure of the content, to achieve a fluid and clear presentation of the results. The Figures will need to flow in parallel with the main text, which will require revising these and improving the presentation, paying additional attention to the legends (which should be more self-contained).

The referees (especially rev #1 and #2) also raise some technical issues that will need to be carefully addressed: rev #1 on modularity, and rev #2 regarding recent issues and criticism of inferences of interactions from simple co-occurrence data.

In light of the reviews (below), we will not be able to accept the current version of the manuscript, but we would welcome re-submission of a much-revised version that takes into account the reviewers' comments. We cannot make any decision about publication until we have seen the revised manuscript and your response to the reviewers' comments. Your revised manuscript is also likely to be sent for further evaluation by the reviewers.

We expect to receive your revised manuscript within 3 months. 

**IMPORTANT - SUBMITTING YOUR REVISION**

*Re-submission Checklist*

*Published Peer Review*

*PLOS Data Policy*

*Blot and Gel Data Policy*

Sincerely,

Roli Roberts

Roland Roberts

Senior Editor

PLOS Biology

rroberts@plos.org

REVIEWERS' COMMENTS:

Reviewer #1:

The authors present here a series of analyses of presence/absence data for 124 taxa across 86 Ediacaran localities to examine whether a large drop in diversity in the late Ediacaran was due to an external disturbance (mass extinction) or the results of eco/evolutionary forces such as niche specialization and competition restructuring the communities. The authors claim that their analyses point to the latter, and while I don't have strong reason to doubt this claim, the conclusions that they draw from their analyses are very hard to follow. In large part this is due to the fact that they a) don't adequately describe the methods they use, and b) don't clearly point the reader to how changes in one metric or another leads to different interpretations and why this should be the case. In part this can be helped by porting some of the basic explanations of what is being done and why to the fore of the manuscript, with details in the methods of course, but even the methods are lacking in terms of specifics. While I understand that the analyses were created and reported elsewhere, I think it is important for the basics to be reported here - especially for the 3 metrics used to parse potential community organizations that they are examining. Apart from the minimal methods, I found the objectives of the paper to be hard to follow. Really what the manuscript needs most is motivation. Why is it important to examine potential changes in community structure, and why are the methods employed the appropriate ones to use in this case? For example - the authors state on L58 "we will investigate the structuring mechanisms within these assemblages". While structure is clearly assessed, it is much less clear how the authors extract *mechanism* from differences in measured structural attributes. While the authors may be quite justified in their interpretation of mechanism, it needs to be clearly stated, which it currently is not. As it stands there are so many unclear aspects of the manuscript, I find it impossible to judge in terms of its technical merits.

My interpretation then is that this paper has a lot of potential - it is an interesting look at a very old community (well, communities). The conclusions that the authors describe are also very interesting as they suggest that a previously interpreted mass extinction may in fact be due to expected changes in the eco/evolutionary trajectories of these communities without such a dramatic externality such as a mass extinction to bring about community change. The pieces are all in place, but the authors need to spend a little more effort to connect them, in my humble opinion. I'd be excited to review an revision that incorporates some of these - mainly textual rather than technical - changes. I was also a bit surprised at the number of grammatical errors and incomplete sentences throughout the manuscript, as if things were quickly shuffled around prior to submission. I'd advise the authors to make make sure that all of their sentences follow grammatical rules during revision and that sentence structure is complete. On a more minor note, ensure that American/British spellings are corrected - both 'specialisation' and 'specialization' are used. Not sure which Plos prefers.

L19 - The authors might consider making it more clear that this is a reinterpretation of the 'diversity drop' described in the first sentence of the abstract, rather than an independent (and not necessarily mutually exclusive) finding.

L21 - Moreover, I think that it is more correct to say that this 'diversity drop' can be explained by an alternative hypothesis (structuring and resulting competitive exclusion, etc)... it is not certain it is one or the other (mass extinction), correct?

L51-53 - I'm not sure what 'using modularity to compare' means... modularity is a network property, but it may be more correct to say that they found differences in the compartmentalization of XXX in the networks rather than to say they use the metric to compare (as it's not a robust statistic for directly comparing networks on its own). Also it is a bit jargony the way it is used here, so some clarification would help. The term 'cluster' used on L53 is also not clear in the way it is used, as it is stated it is a prior analysis, but does this mean it is the same analysis as the one that employs modularity? Overall, the description of the prior work - both the means and the primary conclusions - need to be clarified. It is also important to specify what network is being used. I assume co-occurrence?

L59 - A detail, but 'using a matrix' is less clear than the fact that the authors compiled presence/absence for 86 localities, 124 taxa (wow!)... Whether or not you put this information into a matrix format is less material... If this matrix format is important for the EMS framework, I would suppose it's organization is also important, so should be described a bit more clearly (there are many ways this information can be put into a matrix). 

L67-68 - these metrics need to be defined. They can mean different things in different contexts, and it should be assumed that the reader is not familiar with the EMS framework.

L69 - what does 'fits within the EMS framework' mean? Is there a statistical measure whereby these metrics tell us whether the inputted species across localities are functioning as a spatial meta community? It would be more helpful - rather than to refer to the 'EMS framework' - to tell the reader what the methods employed are telling us ecologically. In other words, I care less about the framework than I care about what the framework tells us regarding the ecology, so the ecology should be put center stage, with the framework being a means to an end. 

Figure 1 - Cool figure, and I think I get the descriptions of the outcomes, but the mechanisms by which these outcomes supposedly derive are completely mysterious. Also - I can't seem to find what NS means… Non-structured? Another very odd thing about this figure is that - for example - coherence is split from left to right as (-) (+) (0)… the other metrics are presented in various order from left to right. Because these metrics are continuous variables, this swapping around in various ways is confusing. If there is a way to order the resulting community structures in a way that makes sense with respect to the continuous nature of the variables?

L70-72 - Can you describe what you mean by random or non-random? There are many ways by which species could be randomly distributed across sites. In the case of this test, what does random mean. More importantly, what does random and non-random imply eco/evolutionarily. The reader needs to be pointed to what these different conditions mean in terms of our interpretation of the community.

L72 - what is reciprocal averaging? Averaging of what?

L67-74 - Both in the abstract and later in the paper, a central point appears to be posing eco/evo restructuring of the meta community as an alternative hypothesis for late Ediacaran diversity declines. If that is a central aim, I would think it would be mentioned here? Overall I think the aims need to be motivated by how they will be filling in knowledge gaps… i.e. you are listing what you intend to do, but not 'why'. 

Regarding this and prior comments on clarity, I understand that the details are in the Materials and Methods, but the Intro/Results/Discussion shouldn't be totally mysterious and jargony. The material needs to be introduced and described in enough detail for the reader to understand why things are being done and what this will mean, even if they will later need to dig into the methods to get the details. So far, I am helplessly adrift as I read through the intro, and I don't know why things are being done and for what reason. Apart from an illusion in the abstract, I'm not even clear that you are testing an alternative hypothesis for the decline in late Ediacaran diversity. The Intro really needs to be re-envisioned to emphasize the aims of the paper, and the motivation for applying the analyses that are described... and what this can perspective can shed light on. [Circling back after reading the materials and methods, they give me a sense of what the measures are measuring, but I don't have a sense of the details. Things like z-scores, which are reported, are not mentioned].

L93-96 - This is a cool finding!

Table 1 - It would be very helpful to orient the readers - in a very basic way - how changes in the three metrics are used to identify the different listed interpretations... i.e. High coherence and low turnover and clumping lead to XXX communities... Low coherence and high turnover lead to YYY communities, etc. Also need to define in Layman's terms what coherence/turnover/clumping means in this context. Not dissimilar from what is written on L155-157, but as a roadmap for the different potential end-conditions in Figure 1. [Circling back, this is done a bit in materials and methods - some of this needs to be placed in the fore, so that the reader can follow what changes in these metrics represent in terms of community structure]

Figure 3 - I can find the (general) definition for boundary clumping (though I think all metrics: coherence, turnover, boundary clumping should be quantitatively defined here - otherwise I have no way of understanding how or why different sites have the values they do), but I have no idea what a 'clump' is, as 1, 3, 6 clumps are reported in Figure 3. Even more problematically, nowhere in the manuscript can I find reference to the z-value displayed in the figure. Because only the raw values of coherence, turnover, and boundary clumping are discussed… with important meanings for (+,-,0), it would be helpful to report why it is preferable to look at the z-values (presumably z-scores?). For the z-scores, what is serving as the mean and sd used to compute values for individual sites? In other words, is the Avalon z score: (Avalon x - Avalon mean) / Avalon sd? Or is it (Avalon x - Total mean)/(Total sd)?

L142 - Wait... what are the 12 models? Either I missed it or it is not mentioned prior to this reference, which is confusing and a bit disconcerting.

L168-171 - Not to be a stats snob, but is this significantly different in a statistical sense? Not everything needs a p-value, but you might consider a different adjective if you use significantly different in a statistical sense elsewhere.

L171 - I think this is a stretch... you don't have interaction information, just co-occurrence information which may or may not (often not) correlate with interactions. I think what is needed here is an argument that the occurrence in different communities implies a change in interactions between species. I would think you could only truly support this if species A is in a completely unique environment where there are no other co-occurring species between the communities being compared. Then species A must be interacting with completely different species in each site. If there are even a small number of co-occurring species in 2 communities with species A, species A might not be changing its behavior - it may just have stronger interactions with the subset of species that co-occur in both sites. So some consideration of the co-occurrence vs. interaction concepts is important here.

L197 - very interesting result!

Figure 4 - this is such a cool figure

L222 - A period missing between 'depth' and 'Claudina'?

L240 - Seems like not a full sentence - something appears to be missing as the next sentence discusses a 'final radiation' but no context wrt earlier radiations.

L247 - Is complexity here being used as a synonym for high community heterogeneity between environments?

L256 - Because this is all inference, I would recommend connecting this inferred mechanism to the pattern that your data bear out...

L269 - the use of 'autecological' and 'synecological' strategies/interactions is used only once. I would suggest using less jargony terms or at least define them. Seems unnecessary given the terms don't appear anywhere else.

L282 - is Turnover capitalized for a reason?

L336 - would checkerboard patterns be random if there is mutual exclusivity? Seems not possible. I would expect more 'randomness' when coherence = 0? I.e. no significant structuring, which implies non-structuring, which I imagine here as somewhat random?

L342 - This implies that Turnover is only defined for coherence > 0? Is that right?

L349 - I think how 'significance' is calculated in each of these contexts needs to be described here.

L351-361 - some of these definitions and verbiage would be much more helpful in the fore-part of the manuscript.

L363 - This is really a personal decision, but because you present 3 primary metrics that form the heart of your manuscript, I would at least consider reporting how they are calculated in the materials and methods unless it is unduly complicated. This would help 'self-contain' the manuscript a bit more and make everything much easier to understand without having to bounce to other publications to gain a deeper understanding of what is going on. 

References - A lot of output that is not in journal format - need curated.

Reviewer #2:

Metapopulation and metacommunity concepts have long been recognized by paleontologists (see, for example, papers in M. McKinney and J. Drake (eds.), 1998. Biodiversity Dynamics). In many ways they are an excellent model for paleontological "communities," which generally can be both spatially and temporally averaged and thus probably represent samples from a regional species pool, rather than a snapshot of a single living community. In this paper, the authors apply the metacommunity concept to Ediacaran assemblages across a wide spatial, temporal, and geographic scale. In particular, they apply the "Elements of Metacommunity Structure" framework, using an R package to analyze the data. There results suggest that the various Ediacaran assemblage show a range of metacommunity structures and are suggestive of an increase in niche specialization over time.

Overall I enjoyed reading this; I have not read about metacommunities in some time and was very pleased to be reintroduced to the concept and to see that it being actively used within paleontology. As far as I know, the use of the Elements of Metacommunity Structure and the mathematical applications of it are novel in paleontology; certainly to the study of the enigmatic Ediacaran assemblages. The figures are very nicely done. There are, however, some questions and suggestions I have before I could recommend publication:

1. Leibold et al. 2004 define "a metacommunity as a set of local communities that are linked by dispersal of multiple potentially interacting species." Given that that the Ediacaran sites within each defined assemblage ("community') are widely separated in space and time, the authors need to discuss how the metacommunity concept applies in this context. Why is it valid to treat these as metacommunities, since there is no evidence they are linked by dispersal? 

2. Figure 1, and much of the results section, make no sense unless one first jumps to the Materials and Methods. 

Although I understand that the journal requires that this section come at the end, that is not necessary for the discussion of the theoretical framework. The material on lines 328-361 needs to be merged with the Introduction section. The description of the data and of the R package can be kept in Materials and Methods. While this is being done, a clearer explanation of the concept of "clumped species loss" is needed; I read line 135-137 several times and I was still left confused (needed to return to the original cited papers). Is that different from "nested species loss" (caption to figure 3)?

3. In Figure 1, the central region with "start" needs to be explained in the caption. This also seems to be redrawn from the Presley et al. (2010) paper, which needs to be cited here,

4. A small point: lines 110-111, 115 "significant p-values highlighted in bold." Significant at what level? To be frank I think simply giving the p-value is sufficient; there are a great many publications that suggest we move away from the concept of "significant." 

5. Lines 129-130: although there are more positive than negative associations, they are both far exceeded by the number of non-associations. This needs to be discussed.

6. Line 173-174. There is no discussion of taphonomic differences among the Ediacaran sites. This is a critical issue that needs to be considered. 

7. Line 240. An incomplete sentence.

Reviewer #3:

I apologize for my delay in reviewing the manuscript due to personal issues. I read with interest the manuscript, which explores a key problem in the interface of macroevolution and ecology. Below, I list my comments on points that I think the authors need to address to improve their manuscript.

My comments:

1. Although the introduction is well written I missed a clear statement of the hypothesis and predictions the authors are testing in this manuscript. For example, Figure 1 describes what I think are hypotheses about metacommunity organization. Having said that, these hypotheses are not proper described in the introduction - the authors only describe what they want to measure but not how these measurements will be used to test predictions of distinct hypothesis.

2. On Figure 1: The laconic legend to figure makes very hard to follow it. This figure could be the heart of the manuscript but its complexity, associated with the minimalist legend imperil its usefulness for the current draft. 

3. The estimating of interactions by co-occurrences is a critical assumption of the manuscript. Recently, the work by Dominique Gravel and others criticized the use of co-occurrences to estimate patterns of interaction. Although I understand the limitations of estimating patterns of interaction in fossil record, I think the authors need to at least discuss these limitations in their manuscript.

4. The use of concepts. There are a few places in the manuscript that the use of the language could be more precise and accurate. For example, in the abstract middle metacommunity may create confusion between middle referring to space or time. Accordingly, in figure 1, stable communities to the description of gleasonian metacommunities may imply these communities are less stable, what is not necessarily true. These communities are just the outcome of species-specific responses to environmental gradients. Please consider to double check the use of these terms in the manuscript.

5. Almost all positive pairwise associations are within assemblages and all negative pairwise associations are between assemblages. Assemblages, in turn, are associated with environmental factors. Thus, I would conclude that there is no need to invoke ecological interactions in shaping patterns of co-occurrence. Or, alternatively, that it is impossible to detangle the effects of environmental gradient from the effects of competition limiting occurrence of species to competitive refugia - since it is of course impossible to perform manipulative experiments. So, I think I am missing something when authors mention the role of competition structuring Ediacarian metacommunities

6. Please consider defining assemblage, community, and metacommunity in our manuscript to avoid confusion.

7. There are some descriptions of tables and figures (e.g., lines 108-111) that should be moved to legend of the table/figure.

8. Lines 142-153: Similar patterns would be predicted by multiple ecological processes. For example, Connell's intermediate perturbation hypothesis led to similar predictions. Similarly, top-down control by predators would lead to lower diversity in environments with both low and very high abundance of predators. So, my question is: is there additional evidence that this is truly a consequence of ecological succession?

9. Lines 165-167. I think I did not follow the reasoning here. If I got it then the approach used here implies that is impossible to record a positive association within this assemblage because the null hypothesis assumes 32/32 positive associations. Is that correct?

10. Line 174-179: I did not follow this flow of ideas about ecological interactions.

11. Line 240. There is something missing in this sentence, I think.

12. The discussion could be improved. There are entire paragraphs with no or just some introductory references, and I missed a truly synthesis with the literature of metacommunities, both from an evolutionary perspective (e.g., Thompson's Geographic Mosaic of Coevolution and Urban and Skelly 2006's evolving metacommunties) and from an ecological perspective (e.g., Guimaraes 2020's Annual review on ecological networks at different scales and Leibold's work on ecological processes in metacommunities).

---

## [Decision Letter · Decision Letter 2]

12 Jan 2022

Dear Dr Mitchell,

Thank you for submitting a revised version of your manuscript "Metacommunity analyses show increase in ecological specialisation throughout the Ediacaran" for consideration as a Research Article at PLOS Biology. This revised version of your manuscript has been evaluated by the PLOS Biology editors, the Academic Editor and the original reviewers. Please accept my apologies for the delay incurred while we experienced difficulties communicating with the Academic Editor over the holiday season.

IMPORTANT: You will see that while the reviewers find the manuscript somewhat improved, they still raise a significant number of concerns that must be addressed. I note a tone of exasperation in their reviews, so please do take this opportunity to make a conscientious effort to thoroughly revise the manuscript and to address these issues; we will only consult the reviewers one more time, and if they still remain dissatisfied then we may decide not to consider the manuscript further. The Academic Editor asked me to emphasise the need for sensitivity analysis: "Regarding the need for some type of sensitivity analysis, I fully agree with rev #1. The sensitivity analysis is not complex and certainly will add much more robustness to the analysis and presentation. The problem is that, somehow, the potentially confounding effects of size difference among the assemblages need to be accounted for. Rev #1 gives some cues about how to frame this."

In light of the reviews (below), we are pleased to offer you the opportunity to address the remaining points from the reviewers in a revised version that we anticipate should not take you very long. We will then assess your revised manuscript and your response to the reviewers' comments and we may consult the reviewers again.

We expect to receive your revised manuscript within 1 month.

**IMPORTANT - SUBMITTING YOUR REVISION**

*Resubmission Checklist*

*Published Peer Review*

*PLOS Data Policy*

*Blot and Gel Data Policy*

Sincerely,

Roli

Roland Roberts

Senior Editor

PLOS Biology

rroberts@plos.org

REVIEWERS' COMMENTS:

Reviewer #1:

General

This revision adds a lot of clarity to the initial submission, however I still have a lot of questions that I do not believe are addressed in the current version. There is still some language confusing taxa interactions vs. taxa co-occurrences that need to be addressed, and I have also tried to note areas where the text can be improved to clarify the motivation of the contribution. Specifically, I found the section detailing the author's hypothesis testing framework to be inadequate and have tried to offer suggestions on that front.

Throughout, and especially in the results, the writing needs work. There are still sentences that aren't following grammatical rules, which I feel - at this level - should have been addressed. I think that the results section could also be better organized to translate the central findings of the paper… perhaps via subheadings.

Unless I missed it completely, one outstanding question that I have with respect to the analysis as a whole is the effect of community size on the metacommunity analysis. Some of the communities vary a lot in size… Nama is very small; White Sea is very large. Are the analyses sensitive to size? If so, which are, and which are not? It's not really addressed (again, unless I missed it), except to say that some communities were not included because they were *too* small (4)… but what is too small? 

Because the difference between the largest (White Sea) and smallest (Nama) is what forms the basis for interpreting whether there was either an extinction or restructuring, understanding what of the results might be influenced by the size difference is *incredibly important* in my view. Incorporating a sensitivity analysis to show that the results are insensitive to community size (hopefully) would be straightforward to do, and I think necessary to disentangle these potential effects. Or perhaps there is a) prior exploration of these effects in previous pubs using this analysis, or b) that White Sea is broken into middle/outer shelves makes the size comparisons more comparable. Yet if the latter is the argument, I would want to understand — looking at Fig 3a — what drives the differences between White Sea (all together) vs. White Sea Middle Shelf, White Sea Outer Shelf, White Sea Deep Subtidal (which look more similar to Nama)… the structure or the size? 

Specific

L13 - It isn't clear in the abstract that assessing these changes in diversity using an 'Elements of Community Structure' framework is posing an alternative hypothesis to a catastrophic mass extinction

L16 - Again, the data collected cannot say anything about how taxa interact… only how they co-occur

L79-81 - is there an assumption here related to even dispersal across all localities? If there was uneven dispersal (some localities connected more strongly than others), would this change any of the assumptions in the analyses?

L84 - it should be mentioned here that these metrics are hierarchical rather than independent

L86 - From Figure 1, I only count 9… I get that the others are the Quasi/Mixed structures, but it is hard to glean from the figure. More generally I don't really understand the quasi structures…

Figure 1 - I find the orientation hard to follow… aside from the (-)(+)(0) layout (I'll let that one go), on the left negative boundary clumping is on top and positive boundary clumping is below; on the right, positive boundary clumping is on top, and negative boundary clumping is below. For the reader to keep all of these attributes in their heads, these stylistic choices can make the paper much harder to follow. There is also an errant 'minus sign' just below 'TURNOVER'… not sure if it's supposed to be there or not?

Intro - this is a nice layout - I like how the first aim is very community structure oriented, the second aim is community v environment, and the third aim is species pairwise correlations… it might be worthwhile to emphasize this tiered approach to the analysis, as I see it as a core strength of this contribution (just a suggestion)

L149 - a strange sentence: "because co-occurence does not necessarily correspond to interaction, here we interpret pair-wise interaction…" do you mean pair-wise correlations?

L153-159 - This section is strangely worded… it is stated that 3 hypotheses will be tested, but proposing a hypothesis test is different than guessing what outcome will occur from an analysis. For example, the current statement goes like this: you 1) hypothesize that increased taxonomic diversity is reflected in terms of increased taxa co-occurrences, 2) hypothesize that Ediacaran data exhibits strong metacommunity structure, and 3) will test whether there is evidence of an extinction… But in reality, you are attempting to *build support* for an alternative hypothesis for extinction… To be more rigorous about this, it seems to me that you are testing between 3 hypotheses:

1) Null: nothing happened (or the signal can't be distinguished from random)

2) Alt Hypothesis 1) an externally-caused mass extinction happened

3) Alt Hypothesis 2) an internally-caused restructuring happened.

You are proposing that a) increased ecological complexity via co-occurrences and b) strong metacommunity structure will lend support to Alt Hypothesis 3, and against 1 and 2 (I presume the Null has already been refuted in prior work showing the diversity drop, etc). Alternatively, if your metacommunity structure analysis supports negative Turnover, that would be support for Alt Hypothesis 2. If I am accurately describing the intent of this paper, it is not very well captured by the organization in Lines 153-160, so I would suggest being a bit more clear about what alternative hypotheses you are testing against, rather than stating guesses at what the results will be.

L177 - I would explicitly state what you are pooling… here it is presence/absence data of organisms as a function of locality. 

L220 - I don't think the z-score is a measure of statistical significance… it is standardized deviance away from the mean… so it is telling us how similar or different each community is from the average across communities in units of standard deviation (i.e. 1 standard deviation away from the mean, etc), if I am understanding it correctly. Echoing my comment in the first draft, this needs to be clearly explained.

L228 - should this be referring to Fig 3? I don't get any insight to Coherence/Boundary Clumping from looking at Fig 2

Generally - what is the effect of size differences between communities? The Nama community is very small and the White Sea community is very big. Are measures of Metacommunity Structure sensitive to size heterogeneity? If so, how could this skew your perspective of these systems? One way to examine this… if you took random subsamples of taxa from the White Sea, where the number of subsamples is equal to that of the Nama community, measured metacommunity structure, and repeated this many many times, do you get measurement distributions with a mean similar to that using the full dataset?

L306 - strangely worded… it begins as a question but without a '?' ends as a statement.

The Results section could really use some editing… there are still quite a few sentences that are poorly constructed, and it overall feels meandering. I had a hard time understanding what the important results were relative to the less-important findings. Easy to get lost in this section. Perhaps some reorganization would help with subheadings? There was also a lot of discussion material in the results section, which I don't personally mind, though it made a discussion section feel superfluous.

L397 - Again (and again), these results cannot say anything about 'interactions between taxa'… only associations

L423-424 - This is a big leap. While there is definitely work showing that generalists may have a selective advantage in the face of large extinction events, it is quite a thing to say that the presence of generalists would imply a catastrophic extinction, or that their absence is incongruent with an extinction!

Reviewer #2:

[IMPORTANT: See attached Word file for fully formatted version, including a Table]

As a reviewer of the original submission, I am glad to have had the opportunity to provide a review of the revised manuscript of this paper. I thought the original paper had promise, but with numerous structural issues, which the other reviewers also recognized. I was pleased to see that they have attempted to address all the issues raised by the reviewers. The current manuscript is a marked improvement as a result. 

That said, the current version still has numerous issues that will require at least one more round of revisions – some of these should have been caught by a careful re-reading prior to resubmission- so that is disappointing. I will try to enumerate these by line. I will also make a suggestion that I hope will improve the clarity of the paper.

1. Line 44. Is “the reduction in taxonomic diversity” different from “diversity drop”? Not clear!

2. Lines 73, 89. Both start with “First.” Reading further, the one on line 89 should be deleted.

3. Lines 73-87. Although there is evidence for larval transport and low provinciality, that does not indicate to me that these can all be treated as a metacommunity in the sense that ecologists would use it. Instead, acknowledge this but point out that for the purpose of this paper we can treat them as metacommunities. 

4. Line 86-127.

a. I went back to the original paper by Presley et al. (ref. 30) and found that there were actually 14 metacommunity types, if one includes Checkerboard and Random. Each of the six categories with positive coherence and significant positive or negative turnover has a non-significant, quasi- equivalent.

b. Even with the revised figure caption and text, I was still confused about the relationships among the metrics and the community types. I was especially confused by what was meant by a “quasi-“ structure and what it implied about the metacommunity. Given that many of the analyses were consistent with a “Clementsian quasi-structure,” this needed further discussion.

c. I have made a table (see below) because it clarified for me the properties of the various metacommunity types. I suggest using something like this in the next revision.

d. I share with one of the other reviewers the issue of capitalization of the metrics. I advise not capitalizing them after they are introduced, and leave the capitalization to the metacommunity types, which are really what the paper should be about. As a comparison, “mean” and “standard deviation” are not capitalized. 

e. Lines 115-124 do not belong here! They are a fragment from the results (lines 304..) (a careful readover should have caught this – do not make this the reviewers job!)

5. Lines 139-142 and Figure 2. Not an accurate description of reciprocal averaging (correspondence analysis). The method ordinates the samples (sites) by their species composition (variable); it also ordinates the species by which site they are in. Variable ordination scores are averages of the case ordination scores and case ordination scores are averages of the variable ordination scores, ”thus “reciprocal averaging.” Both plot on the same axes. Depth is an independent variable, which can then be plotted against the RA axes to help with the interpretation. So, are the axes of Figure 2 are probably the scores for the sites and species on the first RA axis; the left-hand plot of depths is independent data, not used to do the ordination. 

Also: make clear in caption to Figure 2 that this a reciprocal averaging ordination. 

6. Line 172. Again, there are 14, not 12 models, all six illustrated models have quasi- equivalents

7. Lines 177-186. This paragraph badly needs rewriting. 

a. On line 178, you point to Figure 2, but then say nothing at all about it and then jump right away into Figure 3 and then back to Table 1. There needs to be a detailed explanations of both Figure 2 and Table 1, including what is meant by site scores.

b. Looking at the results, it is clear that the relationship between depth and scores only holds true for the Nama, so that it will strongly influence the pooled results. 

8. Lines 188-190. How does the low level of non-random associations compare to what is observed in modern communities or other paleontological examples? Is this unusually low or high? See: LYONS, S. K et al.. 2016. Holocene shifts in the assembly of plant and animal communities implicate human impacts. Nature, 529, 80-83.

9. Line 211-212. What is the simulated mean value of the metric? Not discussed in the text.

10. Lines 256-259. How are the succession stages determined? Needs more detail. 

11. Lines 290, 334 – this is why there needs to be a more detailed explanation of quas-structures!

12. Line 334 – so this is the same structure as the White Sea?

13. Line 395-396. Is this a significant increase, given the large number of random associations? Again, we need context in terms of other communities. 

14. Line 423-426. I am confused here. Based on table 1, the White Sea, except for the outer shelf, and the Nama have the same structure. The change is from the Avalon.

15. Line 468… this is Methods and should be so indicated. 

16. Lines 469-487. This is confusing; both the RA and metacommunity analyses are being discussed in the same paragraph. Break these apart and make sure the RA is carefully described.

 

 Metrics

Community type Properties Coherence Range Turnover Boundary Clumping

Random No structure Random 

Checkerboard High number mutually exclusive pairs; taxa do not respond to gradient Negative 

Nested clumped Species poor communities subsets species rich communities; community synchronous response to gradient Positive Negative Positive

Nested random Species poor communities are random subsets species rich communities Positive Negative Not significant

Nested hyperdispersed Species poor communities subsets species rich communities; species respond individualistically to gradient Positive Negative Negative

Clemenstsian Community synchronous response to gradient Positive Positive Positive

Gleasonian Species respond individualistically to gradient Positive Positive Not significant

Evenly spaced Species spread along gradient with little overlap Positive Positive Negative

Quasi-nested clumped Species poor communities subsets species rich communities; community synchronous response to gradient, fewer turnover than random but not significant Positive Negative but not significant Positive

Quasi-nested hyperdispersed Species poor communities are random subsets species rich communities; fewer replacements than random but not significant Positive Negative but not significant Not significant

Quasi - nested random Species poor communities subsets species rich communities; species respond individualistically to gradient; fewer replacement than random but not significant Positive Negative but not significant Negative

Quasi-Clementsian Community synchronous response to gradient; more replacements than random but not significant Positive Positive but not significant Positive

Quasi- Gleasonian Species respond individualistically to gradient more replacements than random but not significant Positive Positive but not significant Not significant

Quasi-evenly spaced Species spread along gradient with little overlap; more replacements than random but not significant Positive Positive but not significant Negative

---

## [Editor Report · Decision Letter 3]

29 Mar 2022

Dear Dr Mitchell,

On behalf of my colleagues and the Academic Editor, Pedro Jordano, I'm pleased to say that we can in principle accept your Research Article "Metacommunity analyses show an increase in ecological specialisation throughout the Ediacaran Period" for publication in PLOS Biology, provided you address any remaining formatting and reporting issues. These will be detailed in an email that will follow this letter and that you will usually receive within 2-3 business days, during which time no action is required from you. Please note that we will not be able to formally accept your manuscript and schedule it for publication until you have completed any requested changes.

IMPORTANT:

a) You'll see that I've changed your title slightly (inserted "an" and appended "Period") for our wider readership.

b) Thanks you for providing the data and code as supplementary files and in Figshare. We will need one/both of these to be cited as the location of the underlying data in all relevant Figure legends, and I've left a note with one of my colleagues to request this.

Sincerely,

Roli Roberts 

Roland G Roberts, PhD 

Senior Editor 

PLOS Biology

rroberts@plos.org